# Different Cell Types Affect the Transition from Juvenile to Mature Phase in Citrus Plants Regenerated through Somatic Embryogenesis

**DOI:** 10.3390/plants11141811

**Published:** 2022-07-08

**Authors:** Caterina Catalano, Loredana Abbate, Sergio Fatta Del Bosco, Antonio Motisi, Francesco Carimi, Roberto De Michele, Francesco Mercati, Anna Maria D’Onghia, Angela Carra

**Affiliations:** 1CNR—Istituto di Bioscienze e BioRisorse, Corso Calatafimi 414, 90129 Palermo, Italy; caterina.catalano@ibbr.cnr.it (C.C.); loredana.abbate@ibbr.cnr.it (L.A.); sergio.fatta@ibbr.cnr.it (S.F.D.B.); antonio.motisi@ibbr.cnr.it (A.M.); roberto.demichele@ibbr.cnr.it (R.D.M.); francesco.mercati@ibbr.cnr.it (F.M.); angela.carra@ibbr.cnr.it (A.C.); 2CIHEAM—Centre International de Hautes Etudes Agronomiques Méditerranéennes of Bari, Via Ceglie 9, 70010 Valenzano, Italy; donghia@iamb.it

**Keywords:** flowering, juvenile traits, genetic stability, flow cytometry, plant tissue culture, somaclonal variation, thorniness

## Abstract

Robust protocols for the regeneration of somatic embryos in vitro are essential for the efficient use of the most modern biotechnologies. Unfortunately, in perennial trees such as *Citrus*, plants regenerated from juvenile tissues usually exhibit strong, undesirable juvenile characters such as thorny habit and delayed flowering and fruit production. In this work, we tested whether the cell types (nucellar and stigma/style) used to regenerate *Citrus* plants through somatic embryogenesis affected the transition from the juvenile to mature phase. The results show that regenerants from nucellar cells presented persistent juvenile characters, whereas plants originating from stigma/style explants transited to the mature phase more rapidly. Our observations support the hypothesis that the totipotent cells originated from different cell types are not equivalent, possibly by maintaining memory of their previously differentiated state.

## 1. Introduction

The relevance of the *Citrus* industry and the continuous introduction of new improved genotypes encourage the use of biotechnologies based on somatic embryogenesis as an effective tool to rapidly regenerate genotypes of interest. One of the main problems that may limit the use of somatic embryogenesis is the occurrence of somaclonal variation. Plantlets derived from in vitro culture might develop altered characteristics and provide a wide range of culture-induced genetic variants [1] called somaclonal variations [2]. Several factors influence the onset of somaclonal variation, with the type and origin of explant being the most influencing elements [3]. Moreover, plantlets regenerated in vitro through somatic embryogenesis may display ploidy change that induces several anatomical and morphological changes in regenerants [4]. However, the detection of genetic instability in regenerants can be easily addressed through flow cytometric analysis and DNA-based techniques, such as RFLP, RAPD, ISSR, AFLP and microsatellites [5].

Improvement by conventional breeding for *Citrus* is problematic due to several factors such as sterility, nucellar embryony and long juvenile periods [6,7]. The long juvenile period is probably the major constraint for breeders. In *Citrus*, the mature and juvenile forms show distinct morphological characters such as leaf shape, branch habit, growth habit and degree of thorniness. The transition time from juvenile to mature forms varies from species to species [8] and it is also affected by environmental clues [9,10,11]. Early fruit production is a strongly desired character in *Citrus* and, as a consequence, attempts for shortening juvenile period is one of the greatest challenges. There are both historical work and ongoing efforts to use horticultural methods such as hybridization and clonal selection for shortening the juvenile period [12,13,14,15,16,17]. More recently, protocols for genetic transformation aimed to reduce juvenile period have been proposed [18,19,20,21,22].

*Citrus* genetic and sanitary improvement by conventional methods alone has many limitations that can be overcome using in vitro biotechnologies as somatic embryogenesis, which is reported as a key regeneration pathway in many experimental approaches to cultivar improvement [6]. The final success depends on several factors including the age of the explant. In *Citrus*, the best in vitro results, in terms of rapid proliferation rate, are normally obtained using stock material in the juvenile phase as explant source. Sim et al. [23] and Cervera et al. [24] have reported that explants collected from juvenile *Citrus* plants provide the best regeneration frequency in plant tissue culture as compared to explants collected from adult plants. The limited use of adult explants is due to the low morphogenetic potential of explants, and poor rooting of the shoots obtained [25,26]. For this reason, the regeneration of *Citrus* is usually achieved through the culture of nucellar tissues collected from immature, aborted and unfertilized ovules [27]. Unfortunately, plants regenerated from juvenile tissues usually exhibit strong and undesirable juvenile characters for several years. Other subsequent studies indicated the embryogenic potential of somatic tissues which are neither nucellar nor ovular in origin: anthers [28], juice vesicles [29] and stigmas/styles [30]. Among these three different types of explants, stigmas/styles showed the highest embryogenic potential in different *Citrus* species.

In this study, the effects of different cell types (nucellar and stigma/style) on the transition from the juvenile to the mature phase were evaluated on plants regenerated through somatic embryogenesis in four *Citrus* species. Flowering and morphological traits were assessed on plants, grafted onto sour orange, maintained in greenhouse and field conditions. Since DNA and ploidy variation may induce several morphological changes that could influence regenerant growth, the genetic fidelity of the regenerated plants was verified by flow cytometric analysis and DNA analysis (ISSR and RAPD).

## 2. Materials and Methods

### 2.1. Plant Regeneration

Somatic embryos were generated from stigma/style explants dissected from flowers before opening (Figure 1A,B) and undeveloped ovules were dissected from mature fruits (Figure 1C). Plant material of six cultivars belonging to four *Citrus* species (‘Femminello comune’ and ‘Lunario’ lemon (*Citrus limon* (L.) Burman F.), ‘Tardivo di Ciaculli’ mandarin (*Citrus deliciosa* Tenore), ‘AA CNR 31’ sour orange (*Citrus aurantium* L.), ‘Brasilian NL92’ and ‘Valencia late’ sweet orange (*Citrus sinensis* (L.) Osbeck)) was collected from plants growing in the Collesano field station (38° N, 14° E), Sicily. Flowers were surface sterilized by immersion for 5 min in 70% ethanol, 15 min in 2% sodium hypochlorite, followed by three 3 min rinses in sterile distilled water. Stigmas and styles were excised with a scalpel and vertically plated as single explants into medium-sized Petri dishes (100 × 15 mm) with the cut surface in contact with the medium. Mature fruits were harvested 6 months after anthesis. Each fruit was washed, the skin was peeled off and the fruits were surface-sterilized by immersion for 5 min in ethanol (70% *v*/*v*) and 30 min in 2% (*w*/*v*) sodium hypochlorite. Without rinsing, the fruits were cut open under sterile conditions, and the undeveloped ovules were dissected and transferred into medium-sized Petri dishes (100 × 15 mm). Ovule integuments were removed with the aid of a stereo microscope and plated. Explants were cultured on Murashige and Skoog (MS) medium [31] supplemented with 146 mM sucrose, 500 mg L^–1^ malt extract and 13.3 µM 6-benzylaminopurine. The pH of the media was adjusted to 5.7 ± 0.1 with 0.5 M of KOH before autoclaving. Explants and calluses were subcultured into fresh medium at 4–6-week intervals and maintained in a growth chamber at 25 ± 1 °C under a 16 h day length photoperiod. Germinated embryos were isolated and transferred into test tubes (1 embryo per 55 × 23 mm glass tube sealed with Parafilm M) containing 20 mL of the above-mentioned medium. Embryos were considered germinated when there was root extension and hypocotyl elongation. For acclimatization, plantlets (about 3 cm in length) were transplanted into autoclaved Jiffy peat pellets and maintained on a heating bench at 25 °C and at high relative humidity (95%). The conditions for the acclimatization of regenerated plants by grafting have been previously described in De Pasquale et al. [32].

### 2.2. Assessment of Ploidy by Flow Cytometric Analysis

Flow cytometry (FCM) was used to analyse the relative nuclear DNA content of the leaf cells collected either from regenerants and from the relative mother plants used as internal diploid standard (STD 2C). The analysis was carried out with the Partec PAS flow cytometer (Sysmex Partec, Görlitz, Germany, https://www.sysmex-partec.com/; accessed on 18 January 2021), equipped with a mercury lamp. Fully expanded leaves were chopped, using a sharp razor blade, in 400 μL nuclei extraction buffer (solution A of the ‘High Resolution Kit’ for PlantDNA, Sysmex Partec, Germany) for 30–60 s. After filtration through a 30 μm Cell-Trics disposable filter Cell-Trics Sysmex Partec, Germany, 1.6 mL staining solution containing the dye 4,6-diamidino-2-phenylindole (DAPI; solution B of the kit) was added. Routinely, 4000–5000 nuclei were measured per sample and histograms of DNA content were generated using the Partec FlowMax software package.

### 2.3. Assessment of Genetic Stability in Regenerants by ISSR and RAPD Markers

Leaves collected from regenerants and mother plants for each cultivar were harvested, washed, frozen in liquid nitrogen and stored at −80 °C until analyses. Genomic DNA was isolated from the samples as described by [33] and was quantified by measuring OD260 as described by [34]. The isolated genomic DNA was used for ISSR and RAPD analyses in order to assess genetic fidelity as described by Carra et al. [35].

A total of 6 ISSR primers [36] were used to amplify the DNA (Appendix A). The primers were purchased from Life Technologies, Gaithersburg, Md. Each 25 μL amplification reaction consisted of 20 mM Tris-HCl (pH 8.4), 50 mM KCl, 2 mM MgCl_2_, 800 μM dNTP, 0.5 μM of each primer, 1 U of Platinum *Taq* polymerase and 30 ng of template DNA. The amplification was performed under the following cycle program: initial denaturation step for 4 min at 94 °C, followed by 36 cycles at 94 °C for 30 s (denaturation), 48.5–52.0 °C (see Appendix A) for 45 s (annealing) and 72 °C for 120 s (extension), followed by a final extension step at 72 °C for 7 min. A total of 25 µL of each PCR-reaction products were electrophoresed on a 1.5% (*w*/*v*) agarose gel containing 1 × TBE (45 mM Tris-borate, 1 mM EDTA) and 0.5 μg/mL aqueous solution of ethidium bromide. The gel was run for 4 h at 100 V and visualized under UV light lamp. Only those bands showing consistent amplification were considered; smeared and weak bands were excluded from the analysis. Polymorphic ISSR markers were scored for the presence or absence of bands.

RAPD analysis of the grapevine genotypes was performed using six decamer primers [37] (Appendix A). DNA amplification reactions were carried out in a volume of 25 μL with 20 mM Tris-HCl (pH 8.4), 50 mM KCl, 3 mM MgCl_2_, 800 μM dNTP, 0.4 μM of each primer, 1.5 U of Platinum *Taq* polymerase and 25 ng of template DNA. The amplification was performed in a MJ Research thermocycler (Genenco) equipped with a Hot Bonnet under the following cycle program: initial denaturation step for 90 s at 94 °C, followed by 36 cycles at 94 °C for 1 min (denaturation), 36 °C for 60 s (annealing) and 72 °C for 2 min (extension), followed by a final extension step at 72 °C for 10 min. Reaction products were visualized and analyzed as cited for ISSR analysis.

### 2.4. Grafting Conditions

In May, scions generated from different embryogenic events were grafted (T-budding) on three-year-old sour orange rootstocks in greenhouse and field trials, 30 cm above the soil line as described in De Pasquale et al. [32]. As control, to represent the ‘true adult’ state, mature scions collected from mother plants were grafted onto sour orange rootstocks. After the successful graft implant, the upper part of the rootstock was removed.

### 2.5. Growth and Flowering Assessment of Plants Maintained in Greenhouse

Grafted plants were maintained in the greenhouse (six plants regenerated in vitro for each explant/genotype combination and two mother plants for each genotype) in order to check growth, flowering and thorns production. As *Citrus* plants tend to bloom in the uppermost part of the canopy, the lateral branches of the grafted plants were removed, starting from the second year of growth, to induce the scion growth in height as a single leader.

### 2.6. Morphological Analyses on Plants Transplanted in the Field

Grafted plants were maintained in the field (20 plants for each explant/genotype combination and eight mother plants for each genotype) for evaluation of leaf area of the plant and morphological characters (thorn length, thorns/nodes ratio). The experiment used a randomized block design (4 blocks) with 5 replications containing regenerated and mother plants. Leaf area of the plant and morphological analyses were performed on plants growing in the field at the end of the vegetative season (November) for one and three years, respectively.

### 2.7. Estimation of Leaf Area Using Linear Leaf Measurements

Non-destructive methods for measuring the leaf area (LA) through linear measurements have been reported for *Citrus* by many authors [38]. The following criteria for the selection of a regression equation were used: coefficients of determination (*r*^2^), standard errors of estimates, F test of analysis of variance and significance of the regression coefficients (SPSS-X, Inc., Chicago, IL, USA). We found that the leaf area estimation equations had the relative advantage of simplicity of calculation and the lowest standard error of estimates (Table 1).

## 3. Results

### 3.1. Somatic Embryogenesis

Most of the explants produced a creamy-white callus after 1–2 weeks of incubation (Figure 1D). The different genotypes showed a different embryogenic potential from stigma/style and undeveloped ovule explants. About 3–5 months after culture initiation, all of the cultivars regenerated somatic embryos from stigma/style explants (Figure 1E). On the contrary, ‘AA-CNR-31’ sour orange, ‘Valencia late’ sweet orange, ‘Femminello comune’ and ‘Lunario’ lemon generated few somatic embryos from undeveloped ovule explants. Only ‘Tardivo di Ciaculli’ mandarin and ‘Brasiliano NL 92’ sweet orange regenerated a sufficient number of somatic embryos from undeveloped ovules from field and greenhouse trials. About 12 weeks after germination, somatic embryos developed into plantlets (Figure 1F) at a high frequency (40–66%). When plantlets were transferred ex vitro in Jiffy peat pellets, the percentage of acclimatized plants was about 70% (Figure 1G).

### 3.2. Genetic Fidelity Analysis of the Regenerated Plants

Flow cytometric analysis was used to determine the ploidy level of regenerants. All plants regenerated trough somatic embryogenesis of the four different species showed the same ploidy of the mother plants confirming the stability of the ploidy level of plants regenerated from stigma/style and undeveloped ovule explants. Histograms of the DNA content of isolated nuclear suspension of regenerated plants are shown in Appendix A.

Six ISSR and six RAPD primers were screened out and used to amplify 16 DNA samples of regenerants from each cultivar and comparing them to the respective mother plant. A total of 135, 144, 146 and 134 well-resolved band classes were obtained for lemon, mandarin, sour orange and sweet orange, respectively.

The six ISSR primers gave 65, 76, 72 and 66 well-resolved band classes in lemon, mandarin (Appendix A), sour orange and sweet orange. The sizes of amplified fragments were among 250 bp to 3.1 Kb. The mean number of ISSR bands obtained for each primer varied from 7 [primer (TCC)_5_RY] to 15 [primer (AG)_8_YT].

The six RAPD primers produced 70, 68, 74 and 68 well-resolved band classes in lemon, mandarin, sour orange and sweet orange, respectively, ranging from 300 bp to 3.5 Kb in size. The mean number of bands for each primer varied from 8 in ‘Valencia’ with primer OPM 04 (Appendix A) to 17 in sour orange with primer OPAT 14. A total of 14,076 bands (number of plantlets analyzed × number of band classes obtained with all the primers) were generated by the RAPD and ISSR techniques. All regenerated plantlets appeared to be completely identical to the respective mother plants.

### 3.3. Growth and Flowering Assessment of Plants Maintained in Greenhouse

Mother plants and regenerated plants, growing under greenhouse conditions, were inspected for the presence or absence of flowers. The presence of flowers in plants maintained in the greenhouse for three years after grafting is reported in Table 2. During the first year of vegetation, mature and regenerated plants did not flower. During the second year, most mother plants were flowering and only few plants of lemon and sour orange produced flowers and some fruits, while juvenile characters started to be lost on some shoots.

The third year, all the mother plants produced flowers and some of the plants regenerated from stigma/style of lemon, mandarin, sour and sweet orange were flowering under greenhouse condition (Figure 1H). Sour orange (‘AA-CNR-31’) showed the higher percentage of flowering plants (66%), whereas a lower percentage (50%) was observed in lemons (‘Femminello comune’ and ‘Lunario’) and ‘Brasiliano NL 92’ sweet orange. The lowest percentage was detected in ‘Tardivo di Ciaculli’ mandarin and ‘Valencia late’ sweet orange (33%). The third year some of the plants of ‘Femminello comune’ lemon, ‘Brasiliano NL 92’ sweet orange, ‘Tardivo di Ciaculli’ mandarin and ‘AA-CNR-31’ sour orange regenerated by stigma/style produced fruits in greenhouse (Figure 1I,J). Fruits produced by the plants regenerated in vitro from stigma/style did not show differences to those produced by mother plants. Juvenile characters of all genotypes regenerated trough stigma/stile culture started to be lost in several shoots (Figure 1K).

In contrast, none of the plants regenerated from undeveloped ovules (‘Tardivo di Ciaculli’ mandarin and ‘Brasiliano NL 92’ sweet orange) produced flowers in the first three years of greenhouse cultivation and most of the shoots retained their juvenile characters (Table 2).

### 3.4. Observations of Juvenility Mature and Regenerated Plants Maintained in the Field

Plant leaf area and number of thorns per plant, was measured only the first year. Regenerants (stigma/style and undeveloped ovule) showed a higher vegetative growth as compared to the mother plants (Figure 2). In fact, all regenerants had a higher plant leaf area than the mother plants. The lowest plant leaf area was observed in mandarin mother plants (22.3 dm^2^), higher values were observed in stigma/style and undeveloped ovule regenerants: 32.2 and 38.7 dm^2^, respectively. Generally, the plant leaf area of undeveloped ovules regenerants is superior to stigma/style regenerants (Figure 2).

All regenerants exhibited a greater number of thorns as compared to mother plants (Figure 2). Mandarin regenerants from stigma/style and undeveloped ovule explants showed the higher number of thorns (361 and 490, respectively) in the first year, nevertheless also the mother plants showed a high number of thorns (119). In the first year, the lowest number of thorns was observed in sour orange mother plants (9). The different genotypes showed different levels of thorniness: both in regenerated plants, ranging from 75 in sour orange (stigma/style regenerants) to 490 in mandarin (undeveloped ovule regenerants) and in mother plants, ranging from 9 in sour orange to 157 in ‘Lunario’ lemon (Figure 2).

The four species showed great differences in thorn length between mother and regenerated plants (Figure 3).

Regenerated plants after one year of growth in the field showed much longer thorns than mother plants. Moreover, the undeveloped ovule regenerants showed a longer thorn length when compared to the stigma/style regenerants. ‘Brasiliano NL 92’ sweet orange stigma/style and undeveloped ovule regenerants showed the highest thorn length (31.7 and 35.8 mm, respectively) in the first year, yet also the mother plants showed long thorns (17.3 mm).

A similar behavior was observed in mandarin, with stigma/style and undeveloped ovule regenerants showing the highest thorn length (27.4 and 34.5 mm, respectively) as compared to the mother plants (13.2 mm). The lowest thorn length was observed in sour orange, in which stigma/style regenerants had an average thorn length of 10.5 mm and the mother plants thorns were 6.0 mm long. During the second and third year of growth, the thorn length of the regenerated plants was reduced as compared to the first year, but thorns were still longer in regenerated compared with mother plants. The second and third years, plants regenerated from undeveloped ovules produced longer spines than plants regenerated from stigma/style.

Consistent differences were observed in thorn/node ratio between mature and regenerated plants during the three years of growth in the field (Figure 4).

All regenerants exhibited a higher thorn/node ratio as compared to mother plants. Among the plants originating from somatic embryogenesis the undeveloped ovule regenerants showed a higher thorn/node ratio when compared to the stigma/style regenerants. Mandarin regenerants from stigma/style and undeveloped ovule explants showed the highest thorn/node ratio (0.80 and 0.90, respectively) in the first year, while the mother plants showed a lower thorn/node ratio (0.30). In the first year, the lowest thorn/node ratio was observed in sour orange mother plants (0.03). The highest thorn/node ratio of the mother plants was observed in ‘Lunario’ lemon (0.4). During the second and third year of growth, the thorn/node ratio of the regenerated plants was reduced as compared to the first year, with the exception of the mandarin undeveloped ovule regenerants, which maintained similar values in the three years of observation (0.90 and 0.87 for the second and third year, respectively).

At any evaluation time, our observations showed that the thorniness of regenerated plants was higher than those of mother plants; however, some regenerants developed thornless apical shoots. If the thornless shoots were collected and regrafted on sour orange, they kept the acquired mature characters (data not reported). Most juvenile characters (thorniness, internode length, absence of flowers, etc.) were no longer present after mature shoots from plants regenerated in vitro were grafted again and the new vegetation displayed mature morphology nearly identical to mature plants. The only exception was observed in sweet orange and mandarin, which in some cases reverted to the juvenile form. However, in all species, it was possible to obtain plants regenerated from stigma/style explants that displayed mature morphology within three years after grafting. In contrast, undeveloped ovule regenerants retained their juvenile characteristics during the three years of observation after grafting.

## 4. Discussion

Since plants cannot escape adversity, they developed a high regeneration ability for survival to biotic or abiotic stresses. Compared to animals, plants generally have a high level of plasticity and exhibit a remarkable regenerative capacity, both in vivo and in vitro, that varies widely between species and tissue types. Growing evidence suggests that some forms of plant regeneration involve the reprogramming of differentiated somatic cells, while others are induced through the activation of relatively undifferentiated cells in somatic tissues [39]. An extreme example of this adaptation is the generation of adult plants via somatic embryogenesis without the need for fertilization [40]. The regenerative potential can be increased in vitro by exogenously supplied plant growth regulators, wherein the interaction between auxin and cytokinin influences the developmental fate of cells inducing shoot or root regeneration. A balanced concentration of auxin and cytokinin induces an unorganized growth of a cell mass known as ‘callus’ due to its resemblance to the wound-healing plant tissue [41].

Somatic embryogenesis is a powerful biotechnological tool for the propagation and genetic improvement of plants. In *Citrus*, the production of embryogenic callus lines was first reported by Rangan et al. [42] from excised nucelli culture, the regeneration of somatic embryos from stigma and style cultures was first reported by Carimi et al. [43] and, since then, somatic embryogenesis has been induced directly or indirectly via callus formation in several *Citrus* species to produce plants for mass propagation, breeding program [6] and virus-free plants [44,45,46,47,48,49]. Even if somatic embryogenesis is widely used, little information is available on the behavior of *Citrus* plants regenerated in vitro from stigma/style and nucellar culture when grafted on rootstock and transferred under greenhouse conditions.

Here, we tested the genetic fidelity and agronomical aspects of plants regenerated from stigma/style explants or nucellar tissue and grown in the field or greenhouse. Shifts in morphological characters in regenerated plants can be expressed in terms of loss of apical dominance, number and size of leaves and, most importantly, in the time of flowering.

In order to investigate the presence of somaclonal variations on the regenerated plants, we used two different PCR-based techniques, ISSR and RAPD, and flow cytometry to analyse the regenerants. No somaclonal variation compared to the mother plants was obtained, revealing the homogeneity of the produced plantlets. We also observed that fruits produced by the plants regenerated in vitro (from stigma and style cultures) were identical to those produced by mother plants, confirming that regenerated plants are genetically true-to-type. Moreover, this technology is also effective in the elimination of several viral infections [47]. This strategy can be considered as a possible in vitro process for the healthy plant regeneration of *Citrus* with a very low risk of generating somaclonal variants.

Somatic embryo regeneration from mature explants induced a partial reversion to a juvenile state, as widely demonstrated both for woody and herbaceous plants [50,51]. In fact, with the exception of sour orange, most of the regenerated plants showed strong juvenile characters in the first stages of growth, characterized by a high presence of thorns on stems and branches. During the first year of growth (post-grafting), mature and regenerated plants exhibited juvenile growth, characterized by the absence of flowers and high thorniness. As time passed, both mature and regenerated plants reduced these juvenile characters. Most of the lemons and sour oranges had completely thornless single shoots, whereas, in other shoots, thorns were still abundantly present. The loss of juvenility in the early stages of growth confirms previous observations that lemons and sour oranges produced fruits earlier than other *Citrus* species, with fruits present on thornless branches 3 years after the embryogenic event [46]. Similar results were reported for plantlets of Calamondin *Citrus* derived from somatic embryos grafted onto Japanese citron rootstock which, after one year in the field, produced flowers and fruits normally [52]. Propagated plants obtained by regrafting thornless budwood of stigma/style regenerants onto sour orange rootstock did not show any juvenile characters and they were flowering and fruiting regularly. This suggests that the protocol for plantlets regeneration by somatic embryogenesis from stigma/style explants maintains the genomic integrity of the chosen cultivar, reaching the mature stage in a relatively short time.

The loss of juvenile characters proceeded more quickly in mother plants than in regenerated plants. The most obvious explanation is that since the mother plants scions were derived from adult plants, the rejuvenation process was limited to the first year after grafting. Conversely, regenerants by somatic embryogenesis retained the full juvenile potential for a longer period. However, when comparing scions originated by in vitro somatic embryogenesis to scions taken directly from mother plants in the fields, we always need to keep in mind that the somatic embryogenesis process removes most of the viral and endophytic communities of the parent tissue [53]. Therefore, it is possible that the differences in growth and juvenile characters were (also) a result of the interaction of the microbial community in mother plants, versus the “clean” scions derived from somatic embryos.

Most interestingly, we observed a different behavior in plants regenerated from stigma/style versus ovule explants. Regenerants from ovules showed a greater number and density of thorns and they never flowered within the three years of observation. Conversely, regenerants from stigma/style reduced juvenile characters 12–18 months after grafting in the terminal portion of some shoots, and many showed flowers and fruits in the third year. Since both types of regenerants were derived from somatic embryogenesis in vitro, they were originally both free of the viral and endophytic communities. Therefore, the observed differences probably depended on the source of genetic material. Regenerants derived from nucellus, which is a non-vascularized tissue of maternal origin, can be regarded as a pocket of juvenile tissue in an otherwise adult plant [54]. Our results support the hypothesis that the totipotent cells originated from different cell types are not equivalent, possibly by maintaining memory of their previously differentiated state, possibly by epigenetic modifications. Our future research will address the molecular mechanisms underlying the process of juvenility in regenerants.

## Figures and Tables

**Figure 1 plants-11-01811-f001:**
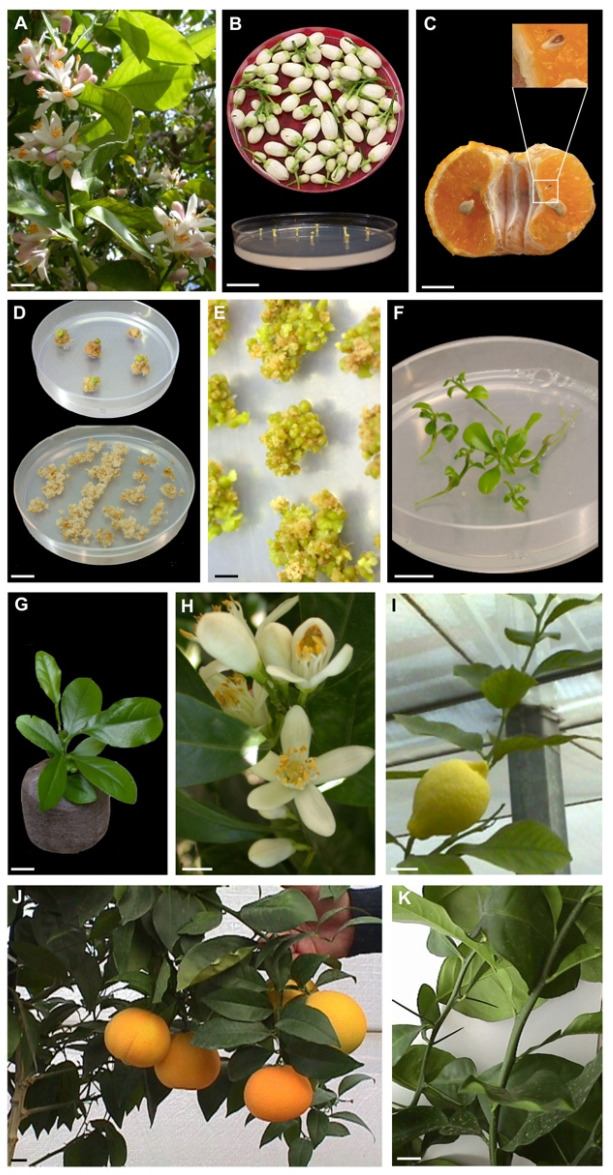
Somatic embryogenesis and plant regeneration in *Citrus*. (**A**) Representative blooming *Citrus* (lemon) (bar = 2 cm); (**B**) Stigma/style explants dissected from orange flowers collected before opening (bar = 2 cm); (**C**) Undeveloped ovule in open pollinated fruit of mandarin harvested 6 months after anthesis (bar = 1 cm); (**D**) Creamy-white callus from the stigma/style and undeveloped ovule explants (bar = 2 cm); (**E**) Somatic embryos generated after 3–5 months of culture initiation at the surface of stigma/style explant-derived callus (bar = 3 mm); (**F**) Germinated somatic embryos growing on MS medium (bar = 1 cm); (**G**) Somatic embryo-derived plant of sweet orange transferred to Jiffy peat pellet (bar = 1 cm); (**H**) Sweet orange stigma/stile regenerants flowering under greenhouse condition (bar = 1 cm); (**I**,**J**) Fruits of ‘Femminello comune’ lemon and ‘Brasiliano NL 92’ sweet orange produced by three years old stigma/style regenerated plants growing in greenhouse (bar = 2 cm); (**K**) Thorny and thornless sour orange shoots from three years old stigma/style regenerants (bar = 2 cm).

**Figure 2 plants-11-01811-f002:**
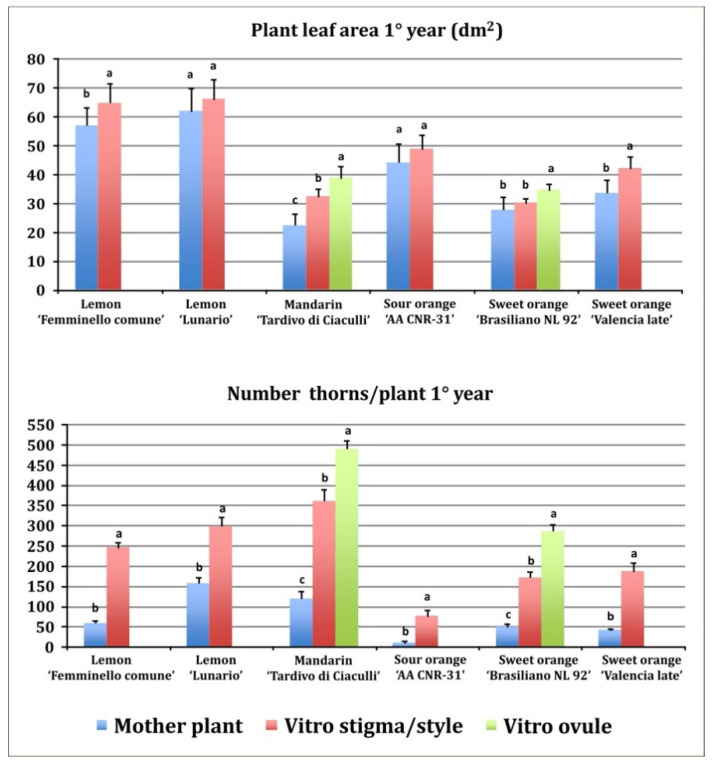
Plant leaf area and number of thorns per plant in the first year of growth in the field. Different letters on bars indicate significantly different values at a particular genotype according to the *t*-test for ‘Femminello comune’, ‘Lunario’ lemon, ‘AA CNR 31’ sour orange and ‘Valencia late’ sweet orange and according to Tukey’s multiple comparison test for ‘Tardivo di Ciaculli’ mandarin and ‘Brasilian NL92’ sweet orange. Tests were performed at *p* < 0.05 significance level. Bars indicate standard error.

**Figure 3 plants-11-01811-f003:**
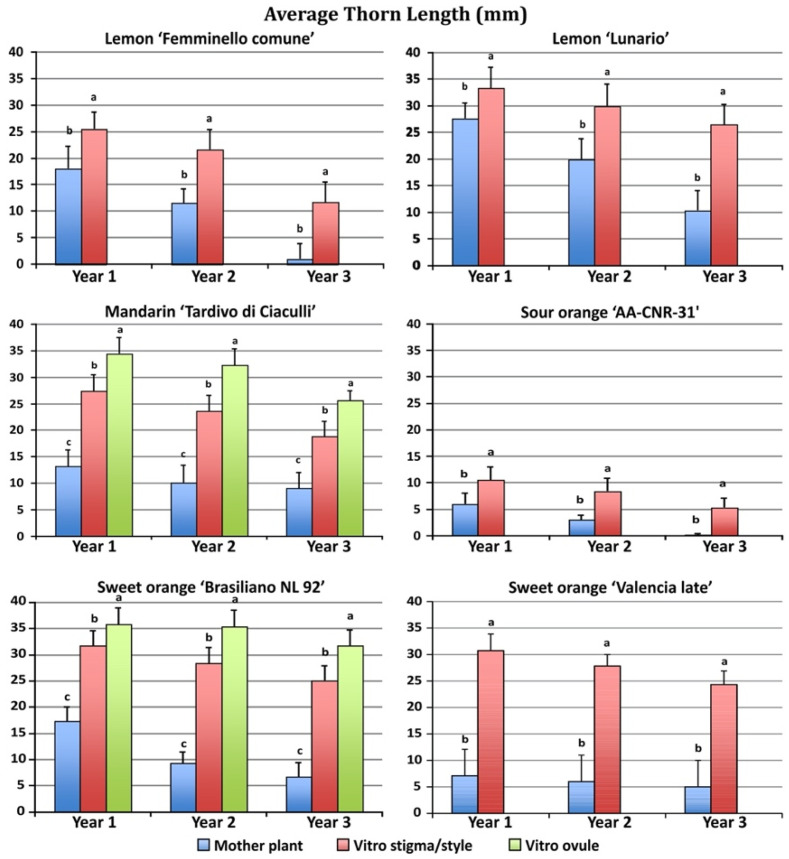
Average thorn length in plants regenerated from different explants growing in the field during the first three years after grafting. Different letters on bars indicate significantly different values at a particular genotype according to the *t*-test for ‘Femminello comune’, ‘Lunario’ lemon, ‘AA CNR 31’ sour orange and ‘Valencia late’ sweet orange and according to Tukey’s multiple comparison test for ‘Tardivo di Ciaculli’ mandarin and ‘Brasilian NL92’ sweet orange. Tests were performed at *p* < 0.05 significance level within each year. Bars indicate standard error.

**Figure 4 plants-11-01811-f004:**
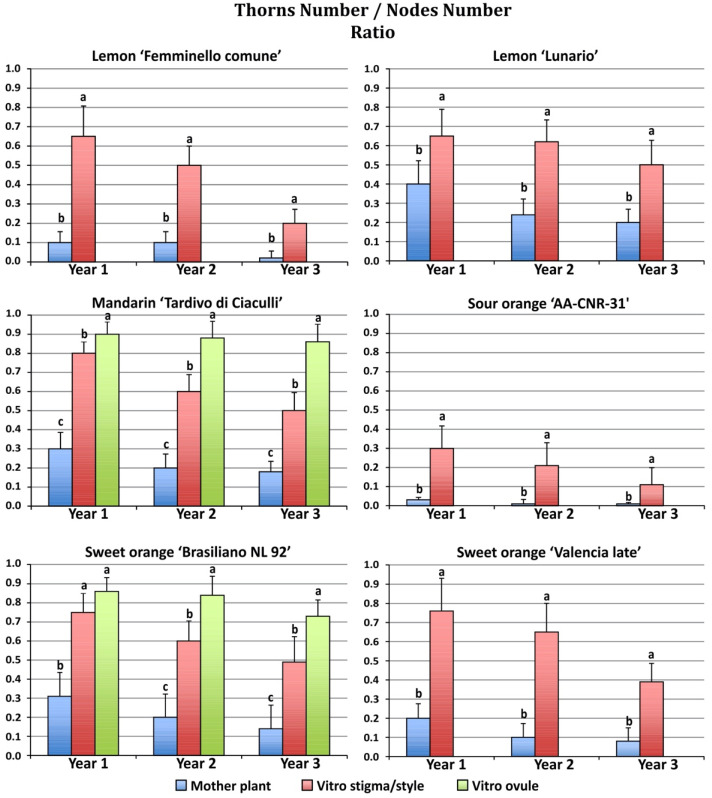
Average thorn/node ratio in plants regenerated from different explants growing in the field during the first three years after grafting. Different letters on bars indicate significantly different values at a particular genotype according to the *t*-test for ‘Femminello comune’, ‘Lunario’ lemon, ‘AA CNR 31’ sour orange and ‘Valencia late’ sweet orange and according to Tukey’s multiple comparison test for ‘Tardivo di Ciaculli’ mandarin and ‘Brasilian NL92’ sweet orange. Tests were performed at *p* < 0.05 significance level within each year. Bars indicate standard error.

**Table 1 plants-11-01811-t001:** Leaf area estimation equations.

Species	Leaf Area Estimation Equation	*r* ^2^
Lemon	LA = 1.477 + 0.652 (L × W)	0.982
Mandarin	LA = 0.784 + 0.618 (L × W)	0.974
Sour Orange	LA = −0.442 + 0.690 (L × W)	0.968
Sweet Orange	LA = 1.506 + 0.632 (L × W)	0.990

L = leaf maximum length, LA = leaf area and W = leaf maximum width.

**Table 2 plants-11-01811-t002:** Observations on flowering plants in greenhouse.

Genotype	Origin	Presence of Flowers			Flowering Plants	Fruiting Plants
		Year 1	Year 2	Year 3	Year 3 (%)	Year 3
Lemon ‘Femminello comune’	Mother plant	No	Yes	Yes	100	Yes
	Vitro stigma/style	No	No	Yes	50	Yes
Lemon ‘Lunario’	Mother plant	No	Yes	Yes	100	Yes
	Vitro stigma/style	No	No	Yes	50	No
Mandarin ‘Tardivo di Ciaculli’	Mother plant	No	Yes	Yes	100	Yes
	Vitro stigma/style	No	No	Yes	33	Yes
	Vitro ovule	No	No	No	0	No
Sour orange ‘AA-CNR-31’	Mother plant	No	Yes	Yes	100	Yes
	Vitro stigma/style	No	No	Yes	66	Yes
Sweet orange ‘Brasiliano NL 92’	Mother plant	No	Yes	Yes	100	Yes
	Vitro stigma/style	No	No	Yes	50	Yes
	Vitro ovule	No	No	No	0	No
Sweet orange ‘Valencia late’	Mother plant	No	Yes	Yes	100	Yes
	Vitro stigma/style	No	No	Yes	33	Yes

## Data Availability

The data presented in this study are openly available in Appendix A here.

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
