# Peer review of "Different Cell Types Affect the Transition from Juvenile to Mature Phase in Citrus Plants Regenerated through Somatic Embryogenesis"

_plants, 2022, doi:10.3390/plants11141811_

Round 1

Reviewer 1 Report

Title: Different cell types affect the transition from juvenile to mature phase in Citrus plants regenerated trough somatic embryogenesis 

The research is well conducted and the paper is properly written, however I appreciate if authors may kindly take into consideration the following changes:

1. Figure 2: Please provide a sharper figure.

2. Figure 3: Please provide the size of the standard marker.

3 Table 1: The coefficient of determination (r^2) values ​​are low except for sweet orange, but are they significant? Authors need an explanation.

Author Response

We appreciate all the comments made by the Reviewer 1. We have addressed the suggestions made on the manuscript.

Referee 1 comment 1: Figure 2: Please provide a sharper figure.

Author’s answer: A sharper figure has been provided (now Fig. S1).

Referee 1 comment 2: Figure 3: Please provide the size of the standard marker.

Author’s answer: The size of standard marker has been added (now Fig. S1).

Referee 1 comment 3: Table 1: The coefficient of determination (r^2) values are low except for sweet orange, but are they significant? Authors need an explanation.

Author’s answer: Although a good fit has an R2 close to 1.0, this number alone cannot determine whether the data points or predictions are biased. It also doesn't tell analysts whether the coefficient of determination value is intrinsically good or bad. It is at the discretion of the user to evaluate the meaning of this correlation, and how it may be applied in the context of future trend analyses.

Overall, the R2 values we observed in the present study vary between 0.990 (Sweet Orange) and 0.968 (Sour Orange). We believe that even the lowest 'goodness of fit' in our study allows us to correctly assess the plant leaf surface. An R2 equal to 0.968 indicates that there is a good correlation between the dependent and independent variables, in fact we expect that 96.8% of the dependent variable is predicted by the independent variables. This means that we could have an error in the estimate of the leaf area equal to 3.2%, a value in our opinion acceptable.

Reviewer 2 Report

This article reports regenerated citrus plants' morphological and flowering characteristics through in vitro somatic embryogenesis from nucellus or stigma/style. The characteristics of juvenility were phased out year by year, however, the degrees of the disappearance of juvenile characters were different depending on the originated tissues for the embryogenesis and citrus species. These data will be a piece of fundamental information to understand the characteristics of rejuvenation (returning to the juvenile phase) in citrus plants in artificial plant regeneration procedures. 

I noticed several insufficiencies in data presentations and text descriptions. The revisions of the below points will improve the reliability of this research article.

Major comments

1.      The conditions for the regeneration of somatic embryos (L.87-88) are important information. Please briefly describe the procedure of somatic embryogenesis in this article.

2.      Figure 2 and Figure 3 are just confirmation results for genetic stability. These data can be moved to the supplemental data. Because the regenerated plantlets were completely identical to the respective mother plants by ISSR and RAPD analysis, the explanation in the main text is enough to understand the genetic background.

3.      Please reconsider and reperform the statistical analysis for Figures 4, 5, and 6 data. The multiple tests (Tukey’s test ) were adopted. However, multiple data are only for Mandarin ‘Tardivo di Ciaculli’ and Sweet Orange ‘Brasilian NL92’, other cultivars’ data are not multiple, and "t-test" may be enough to understand the significance of the differences. I feel strangeness in some parts of different letter indications in each figure (Fig. 4, 5, and 6). and, What is indicated by error bars in the graphs?

4.      Detailed data for flowering and fruiting are required. The data in Table 2 is too simple. We would like to know the actual number of flowers per individual plantlet and the quality data of flowers (perfect or imperfect to develop fruit or fruiting rate). The other data such as plant size (height, trunk diameter) and tissue colors (leaves, branches, flower buds) are preferred to be presented.  

Minor comments

L.85; Is the concentration of 6-benzylaminopurine right? 13.3 mM looks very high concentration for in vitro usage.

L.440-479; The information in these sections is incomplete. These texts are kind of template text (example text).

Author Response

We appreciate all the comments made by the Reviewer 2. We have addressed the suggestions made on the manuscript.

Referee 2 comment 1: The conditions for the regeneration of somatic embryos (L.87-88) are important information. Please briefly describe the procedure of somatic embryogenesis in this article.

Author’s answer: detailed information has been added to explain the whole procedure of somatic embryogenesis

Referee 2 comment 2:  Figure 2 and Figure 3 are just confirmation results for genetic stability. These data can be moved to the supplemental data. Because the regenerated plantlets were completely identical to the respective mother plants by ISSR and RAPD analysis, the explanation in the main text is enough to understand the genetic background.

Author’s answer: Both figures have been moved to supplemental data as suggested.

Referee 2 comment 3: Please reconsider and reperform the statistical analysis for Figures 4, 5, and 6 data. The multiple tests (Tukey’s test) were adopted. However, multiple data are only for Mandarin ‘Tardivo di Ciaculli’ and Sweet Orange ‘Brasilian NL92’, other cultivars’ data are not multiple, and "t-test" may be enough to understand the significance of the differences. I feel strangeness in some parts of different letter indications in each figure (Fig. 4, 5, and 6). and, What is indicated by error bars in the graphs?

Author’s answer: Statistical analyses has been performed according referee’s suggestions. The caption of every table reported information about tests adopted. Explanation of error bars has also been added.

Referee 2 comment 4: Detailed data for flowering and fruiting are required. The data in Table 2 is too simple. We would like to know the actual number of flowers per individual plantlet and the quality data of flowers (perfect or imperfect to develop fruit or fruiting rate). The other data such as plant size (height, trunk diameter) and tissue colours (leaves, branches, flower buds) are preferred to be presented.

Author’s answer: Requested data are not available. When experiments were performed these data have not been registered and now it is impossible to get them. Plants are adult and acquiring this type of data would take about 5 years.

Referee 2 comment 5: L.85; Is the concentration of 6-benzylaminopurine right? 13.3 mM looks very high concentration for in vitro usage.

Author’s answer: It was a misprint. The error has been corrected changing “m” into “µ”.

Referee 2 comment 6: L.440-479; The information in these sections is incomplete. These texts are kind of template text (example text).

Author’s answer: It was a misprint. Needed information have been added.

Round 2

Reviewer 1 Report

The authors have done a satisfactory job at addressing my concerns, which has improved the manuscript. 

Author Response

We appreciate the comment made by the Reviewer 1.

Referee 1 comment 1: The authors have done a satisfactory job at addressing my concerns, which has improved the manuscript.

Author’s answer: We thank the reviewer 1.

Reviewer 2 Report

The manuscript has been improved following the reviewer’s comments and suggestions.

Further minor suggestions are below;

For the statistical results, the lowercase letter “a” should be attached to the highest value in the comparison. The letters “a” to  “c” should be put in the order of highest to lowest.

Some figure layouts seem broken.

Figures 3 and 4 are time-course data. Multiple comparison test (Tuley’s test) can be treated to all data comparisons in each panel (each cultivar)  regardless of the different explants.

Author Response

We appreciate all the comments made by the Reviewer 2. We have addressed the suggestions made on the manuscript.

Referee 2 comment 1: For the statistical results, the lowercase letter “a” should be attached to the highest value in the comparison. The letters “a” to “c” should be put in the order of highest to lowest.

Author’s answer: The order of letter has been changed.

Referee 2 comment 2: Some figure layouts seem broken.

Author’s answer: The problem was solved and layouts are now correct.

Referee 2 comment 3: Figures 3 and 4 are time-course data. Multiple comparison test (Tuley’s test) can be treated to all data comparisons in each panel (each cultivar) regardless of the different explants.

Author’s answer: Thank you for the remark. Our intent was to compare the behavior of plants of the same age in relation to the explant they derived from. For this reason, the captions of the figures have been changed adding “within each year”.